# Antifungal Combination Eye Drops for Fungal Keratitis Treatment

**DOI:** 10.3390/pharmaceutics15010035

**Published:** 2022-12-22

**Authors:** Victoria Díaz-Tomé, Carlos Bendicho-Lavilla, Xurxo García-Otero, Rubén Varela-Fernández, Manuel Martín-Pastor, José Llovo-Taboada, Pilar Alonso-Alonso, Pablo Aguiar, Miguel González-Barcia, Anxo Fernández-Ferreiro, Francisco J. Otero-Espinar

**Affiliations:** 1Department of Pharmacology, Pharmacy and Pharmaceutical Technology, University of Santiago de Compostela (USC), 15705 Santiago de Compostela, Spain; 2Institute of Materials iMATUS, University of Santiago de Compostela (USC), 15706 Santiago de Compostela, Spain; 3Paraquasil Group, University Clinical Hospital, Health Research Institute of Santiago de Compostela (IDIS), 15706 Santiago de Compostela, Spain; 4Clinical Pharmacology Group, University Clinical Hospital, Health Research Institute of Santiago de Compostela (IDIS), 15706 Santiago de Compostela, Spain; 5Molecular Imaging Group, University Clinical Hospital, Health Research Institute of Santiago de Compostela (IDIS), 15706 Santiago de Compostela, Spain; 6Nuclear Magnetic Resonance Unit, Research Infrastructures Area, Universidade de Santiago de Compostela Campus Vida, 15705 Santiago de Compostela, Spain; 7Microbiology Department, University Clinical Hospital Santiago de Compostela (SERGAS), 15706 Santiago de Compostela, Spain; 8Pharmacy Department, University Clinical Hospital Santiago de Compostela (SERGAS), 15706 Santiago de Compostela, Spain

**Keywords:** fungal keratitis, cyclodextrin, natamycin, voriconazole, PET/CT imaging, nuclear magnetic resonance, cyclodextrin aggregates

## Abstract

Fungal keratitis (FK) is a corneal mycotic infection that can lead to vision loss. Furthermore, the severity of FK is aggravated by the emergence of resistant fungal species. There is currently only one FDA-approved formulation for FK treatment forcing hospital pharmacy departments to reformulate intravenous drug preparations with unknown ocular bioavailability and toxicity. In the present study, natamycin/voriconazole formulations were developed and characterized to improve natamycin solubility, permanence, and safety. The solubility of natamycin was studied in the presence of two cyclodextrins: HPβCD and HPγCD. The HPβCD was chosen based on the solubility results. Natamycin/cyclodextrin (HPβCD) inclusion complexes characterization and a competition study between natamycin and voriconazole were conducted by NMR (Nuclear Magnetic Resonance). Based on these results, several eye drops with different polymer compositions were developed and subsequently characterized. Permeability studies suggested that the formulations improved the passage of natamycin through the cornea compared to the commercial formulation Natacyn^®^. The ocular safety of the formulations was determined by BCOP and HET-CAM. The antifungal activity assay demonstrated the ability of our formulations to inhibit the in vitro growth of different fungal species. All these results concluded that the formulations developed in the present study could significantly improve the treatment of FK.

## 1. Introduction

Fungal keratitis (FK) is a corneal mycotic infection mainly caused by corneal trauma with contaminated plants or objects [1,2], the misuse of contact lenses [3,4], prolonged use of topical antibiotics or corticosteroids [5], eye surgeries [6], or the immunocompromised state of the patient [7,8], among others. FK is a severe disease that can lead to vision loss or even the complete loss of the eye. FK is usually caused by yeasts such as *Candida albicans* or filamentous fungi such as *Aspergillus* spp. or *Fusarium* spp., but it can be caused by more than 100 different species [3,8,9].

The prognosis of FK lies in early diagnosis and correct treatment. One of the biggest problems of FK diagnosis is that the patient may be asymptomatic after the trauma, so the diagnosis might be delayed for days or even weeks until the patient suffers some symptom (ocular pain or sensitivity to light, among others) [10]. Furthermore, the non-specific symptoms of FK can lead to erroneous diagnosis and treatment; for this reason, the microbiological diagnosis must be mandatory to choose a suitable treatment.

There is currently only one formulation approved by the Food and Drug Administration (FDA) for the treatment of FK, Natacyn^®^, which is a conventional natamycin suspension. Natamycin penetration through the cornea to the deeper structures of the eye is hindered by its low aqueous solubility and high molecular weight. Therefore, to achieve therapeutic concentrations, Natacyn^®^ is administered every hour, leading to the poor adherence of patients to the treatment.

Natamycin is a polyene drug of amphipathic nature, and it is practically insoluble in water (30–50 mg/L). Natamycin has a broad spectrum of action against filamentous fungi (e.g., *Aspergillus* spp., *Fusarium* spp.) [11] and yeasts (e.g., *Candida albicans*) [12]. However, although in vitro studies showed natamycin to be effective against *Fusarium* spp., this did not translate into favorable clinical outcomes, probably due to its poor penetration into the deeper corneal layers [13].

In contrast, voriconazole (a fluconazole derivative) is a triazole with a broad spectrum against *Aspergillus* spp., *Candida* spp., *Fusarium* spp., *Scedosporium* spp., and *Paecilomyces* spp., among other fungal species [14]. Voriconazole is widely used for the treatment of FK [6,15,16], but there is still no commercial ophthalmic formulation approved by the FDA or the European Medicines Agency (EMA). Only formulations marketed and approved for oral and intravenous routes are available [17,18]. For this reason, hospital pharmacy departments must reformulate voriconazole formulations intended for other administration routes, usually intravenous. These formulations are reconstituted with ophthalmic buffers, but their toxicity, bioavailability, and stability remain unknown in most cases. Moreover, the high nasolacrimal drainage leads to short ocular permanence and to the systemic absorption of the formulation that may trigger side effects.

The severity of FK is aggravated by the emergence of resistant fungal species. Antifungal combination therapy is more useful than monotherapy in antifungal-resistant fungi infections [19]. For this reason, several studies have been conducted to evaluate different antifungals combinations [20] or combinations between antifungals and other drugs [21]. Previous studies have shown that the combination of natamycin and voriconazole may be more effective, showing synergism or an additive effect in certain species such as *Fusarium* spp. [20,22,23].

Ocular formulations must also be designed considering excipients that are safe for ophthalmic administration and that enhance the formulation properties. The use of cyclodextrins might be considered a suitable approach to improve drug solubility [24]. Moreover, according to the EMA, some cyclodextrins, such as 2-hydroxypropyl-β-cyclodextrin (HPβCD), have demonstrated ophthalmic safety, as well as improved drug permanence on the ocular surface and transcorneal permeability [25].

The main goal of this work was to develop a new ophthalmic formulation for the combination of natamycin and voriconazole based on the need to find an effective and non-invasive treatment for FK. Natamycin and voriconazole formulations were characterized by means of solubility, Nuclear Magnetic Resonance (NMR), pH, osmolality, and viscosity studies. The in vitro release was evaluated to assess the release kinetics. Ocular safety was evaluated by two different organotypic cytotoxicity models, Bovine Corneal Opacity and Permeability (BCOP) assay and Hen’s Egg Test—Chorioallantoic Membrane (HET-CAM). Bioavailability properties were evaluated using freshly excised bovine corneas. Ocular permanence was assessed by a corneal mucoadhesiveness test and confirmed by an in vivo ophthalmic permanence assay using Positron Emission Tomography/Computed Tomography imaging (PET/CT imaging). In addition, the antifungal susceptibility was studied by a disc diffusion method.

## 2. Materials and Methods

### 2.1. Materials

Natamycin was purchased from LabNetwork^®^ (Saint Paul, MN, USA); Voriconazole was procured by Normon^®^ (Madrid, Spain); Hyaluronic Acid (HA) (MW 1.4 × 10^6^ Da) was obtained from Acofarma^®^ (Barcelona, Spain); 2-hydroxypropyl-β-cyclodextrin (HPβCD) (Kleptose^®^, 0.65 molar substitution ratio, MW 1399 Da) was obtained from Roquette^®^ Laisa S.A. (Valencia, Spain); 2-hydroxypropyl-γ-cyclodextrin (HPγCD) (0.6 molar substitution ratio, MW 1580 Da) was purchased from Sigma Aldrich^®^ (Darmstadt, Germany); Liquifilm^®^ was obtained from Allergan^®^ Pharmaceuticals Ireland (Mayo, Ireland); Poloxamer (P407) and Polyvinyl alcohol (PVA) (Mw 30,000–70,000 Da) were procured by Sigma Aldrich; and Methyl Cellulose (MC) (1500 cP) was procured by Shin-Etsu (Japan).

### 2.2. Phase Solubility Diagrams

Solubility diagrams of natamycin were obtained according to the Higuchi and Connors methodology [26]. Natamycin complex stability constants were determined using the solubility diagrams.

The solubility assay was based on the addition of an excess of the drug to different solutions with increasing concentrations of two different cyclodextrins, 2-hydroxypropyl-β-cyclodextrin (HPβCD) or 2-hydroxypropyl-ɣ-cyclodextrin (HPγCD). Cyclodextrin solutions were maintained for 7 days in an orbital shaker (VWR^®^) (25 ± 0.5 °C, 200 rpm) to achieve the maximum solubility of natamycin. Afterwards, the resultant solutions were centrifugated (Eppendorf^®^ Centrifuge 5804R) at 12,000 rpm for 30 min and 25 °C. Natamycin concentration was determined by UV-vis spectrophotometry (Agilent^®^ Cary UV 60, λ = 310 nm) after dilution of an aliquot in 0.1 M acetic acid. Each measurement was performed in triplicate.

Solubility diagrams were obtained by representing the concentration of natamycin (mM) against the concentration of cyclodextrins (mM). The slope obtained from the solubility diagrams, the natamycin water solubility (S_0_), and the natamycin solubility in the intercept (S_0 extrap_) were used to calculate the apparent stability constant (K_1:1_ or K_d_) assuming the obtention of 1:1 ratio inclusion complexes. The equations described by Loftsson et al. [27] were used to calculate the complexation efficiency (CE) and the natamycin:cyclodextrin molar ratio (D:CD) values.

### 2.3. Morphological Analysis by Transmission Electron Microscopy (TEM)

The morphological analysis of the particles of the saturated solution of natamycin in 40% (*w*/*v*) HPβCD solution and the saturated solution of natamycin in 40% (*w*/*v*) HPβCD solution and 1% (*w*/*v*) voriconazole solution were evaluated using a JEOL JEM-F200CF-HR microscope (JEOL^®^. Peabody, MA, USA). Samples were placed on copper grids and stained with 2% (*w*/*v*) phosphotungstic acid. Samples were dried and evaluated by TEM observation using an accelerating voltage of 200 kV.

Particle size was measured using Image-Pro Plus Image Analysis Software Version 6.0.0.260 (Media Cybernetics, Inc., Rockville, MD, USA). 

### 2.4. Natamycin Solubility with HPβCD and Different Hydrophilic Polymers

The solubility of natamycin was studied with 20% (*w*/*v*) HPβCD and different ratios of hydrophilic polymers, these being the following: polyvinyl alcohol (PVA), hyaluronic acid (HA), and poloxamer 407 (P407). This study was carried out to assess the differences of natamycin in terms of solubilization efficiency.

Solubility was determined by adding an excess of natamycin to solutions made up of 20% (*w*/*v*) of HPβCD and different concentrations of hydrophilic polymers (see Table 1). The hydrophilic polymer concentrations were chosen based on previous studies [28,29].

### 2.5. Natamycin and Voriconazole Solubility with HPβCD

The solubility of natamycin was studied at different concentrations of HPβCD and a fixed concentration of voriconazole. Three solutions were made at increasing concentrations of HPβCD (20%, 30%, and 40% (*w*/*v*)) and 1% (*w*/*v*) voriconazole. An excess of natamycin was subsequently added to each solution. Similarly, voriconazole solubility was also studied at different concentrations of HPβCD and a fixed concentration of 0.4% (*w*/*v*) natamycin.

The resultant solutions were incubated for 7 days. Afterwards, the solutions were centrifugated (Eppendorf^®^ Centrifuge 5804R) at 12,000 rpm for 30 min and 25 °C. Natamycin concentration was determined by UV-Vis spectrophotometry (Agilent^®^ Cary UV 60 λ = 310 nm) by previously diluting an aliquot in acetic acid 0.1 M. Each measurement was made in triplicate.

### 2.6. Nuclear Magnetic Resonance (NMR) Studies

Liquid-state NMR spectra were conducted at 25 °C on a Bruker NEO 17.6 T spectrometer (proton resonance 750 MHz), equipped with a ^1^H/^13^C/^15^N triple resonance PA-TXI probe and PFG shielded z-gradient that uses 5 mm standard OD tubes. The spectrometer control software was TopSpin^®^ 4.0. The chemical shifts are referenced to the lock deuterium solvent. Spectra were processed and analyzed with Mestrenova^®^ software v14.0 (Mestrelab^®^ Inc., Santiago de Compostela, Spain).

Samples containing natamycin, voriconazole, and/or HPβCD were prepared in 5 mm standard tubes. The exact concentration of the compounds is indicated in each case.

A two-dimensional HSQC multiplicity edited ^1^H-^13^C spectrum was measured (pulse sequence “hsqcedetgpsisp2.4” of the Bruker library) for a sample prepared with 10 mM of natamycin in 0.6 mL of CD^3^OD. The INEPTs transfers were optimized for a nominal value of the scalar coupling 1JCH of 145 Hz. The delay for multiplicity selection was set to 1/(2·1JCH) to detect with the same sign signals of CH_3_ and CH groups and with opposite phase CH_2_ groups. The relaxation delay (d^1^) and the FID acquisition time (at) were 1.6 and 0.112 s, respectively. The spectrum was acquired with 2048 and 160 complex points in the t_2_ and t_1_ dimensions, respectively. The number of scans per t_1_ increment was 16 and the total measurement time was ~1 h.

One-dimensional Saturation Transferred Difference ^1^H spectra (STD) [30,31] were measured for a sample prepared with 10 mM of natamycin and 10 mM of HPβCD in 0.6 mL of D_2_O. The selective saturation consisted of a train of soft gaussian-shaped pulses of 50 ms duration with a 1 ms interpulse delay. This saturation was applied during 2 s at a specific frequency of the ^1^H spectrum and covers a region of the spectrum of ±125 Hz around the chosen frequency (i.e., ±0.17 ppm in a 750 spectrometer). The STD^off^ saturation was applied at 20 ppm. The STD^on^ saturation was applied at the frequency of one specific aromatic proton signal of natamycin and does not affect any of the signals of the cyclodextrin receptor. The STD^on^ and STD^off^ scans were measured in alternate scans and subtracted by the phase cycling providing the subtracted STD^off-on^ spectrum. Two STD spectra were measured by placing the STD^on^ saturation over a specific signal of natamycin at 6.79, 6.05, and 5.91 ppm corresponding, respectively, to protons H^3^, H^17^-H^22^, and H^2^. Each spectrum was acquired in 15 min with 128 scans and a 6.75 s per scan distributed as 2 s of pre-scan delay d_1_, 2 s of STD saturation-time, and 2.75 s of FID acquisition time.

A ^1^H-NMR titration assay was carried out with the one-dimensional ^1^H spectrum (pulse sequence “zg” of the Bruker library) with 32 scans and a relaxation delay (d_1_) of 2 s, and a fid (free-induction decay) acquisition time (aq) of 2.75 s was measured. The titration study was carried out at a constant concentration of voriconazole and HPβCD of 5.62 and 5.56 mM. The concentrations of natamycin explored during the titration were 0, 1.05, 2.10, 3.15, 4.21, 5.26, 6.31, 7.36, 8.41, 9.46, and 10.51 mM.

### 2.7. Preparation of Formulations

Based on the results of the previous sections, formulations with 40% (*w*/*v*) HPβCD, 1% (*w*/*v*) voriconazole, and 0.7% (*w*/*v*) natamycin were prepared.

The concentration of cyclodextrin (HPβCD) required to solubilize 1% (*w*/*v*) of voriconazole was established at 20% (*w*/*v*) in previous studies by using voriconazole solubility diagrams [32].

An aqueous solution (SLV) was prepared by adding 40% (*w*/*v*) HPβCD to MilliQ^®^ water until complete solubilization. Then, 1% (*w*/*v*) voriconazole and 0.7% (*w*/*v*) natamycin were added to the cyclodextrin solutions and subsequently dispersed by magnetic stirring (200 rpm) at room temperature until complete solubilization.

Two types of hydrogels were prepared with the purpose of increasing the permanence of the formulations on the ocular surface: hyaluronic acid hydrogel (AHNV) and polyvinyl alcohol-based hydrogel (Liquifilm^®^) (LNV).

#### 2.7.1. Preparation of the Hyaluronic Acid Hydrogel (AHNV)

The AHNV was prepared based on previous studies [33,34]. An amount of 0.4% (*w*/*v*) HA was added to the aqueous solution (SNV) and dispersed by magnetic stirring (200 rpm) at room temperature until complete solubilization.

#### 2.7.2. Preparation of the Polyvinyl Alcohol-Based (Liquifilm^®^) Hydrogel (LNV)

The LNV was prepared based on previous studies where Liquifilm^®^ was chosen as a vehicle for the preparation of a topical ophthalmic formulation of tacrolimus [35]. In addition, Liquifilm^®^ is used in ophthalmic pharmaceutical compounding for the preparation of antibacterial eye drops in the hospital pharmacy department [36]. An amount of 40% (*w*/*v*) HPβCD was added to 50% of the final volume of Liquifilm^®^ and dispersed under low-intensity magnetic stirring (50 rpm) until its complete solubilization for 12 h at room temperature to avoid bubble formation. Afterwards, the formulation was made up to the final volume, and 1% (*w*/*v*) voriconazole and 0.7% (*w*/*v*) natamycin were added under magnetic stirring (200 rpm) until complete dissolution.

### 2.8. Transparency

The transparency of the formulations was measured by recording the transmittance in a wavelength range from 800 to 200 nm, using a spectrophotometer (Agilent^®^ Cary UV 60). The wavelength range includes the infrared light band (780 nm onwards), the visible light (380 to 780 nm), and the ultraviolet light band (100 to 380 nm) [37]. Maximum transparency is considered when the transmittance values are 100% in the visible light range. Each formulation was measured in triplicate.

### 2.9. Osmolality, pH, and Viscosity Measurements

Osmolality measurements were taken with a Micro-Osmometer (Fiske^®^ Model 210). The pH was measured using a pH meter (HI5221 HANNA^®^) at 25 ± 0.5 °C. Viscosity was tested with a rotational viscosimeter (Visco QC 300 Anton Paar^®^) at 25 °C and 20 rpm. Each determination was carried out in triplicate.

### 2.10. Quantitative Analysis: Ultra-High-Performance Liquid Chromatography (UHPLC)

The concentrations of natamycin and voriconazole were determined using a UPLC Waters HClass Plus (Waters, France) with an FTN injector and PDA detector. The column used was a Waters Acquity BEH C18 (2.1 × 50 mm, 1.7 µm) thermostated at 25 °C. The mobile phase was acetonitrile:ammonium acetate buffer (30:70 *v*/*v*) using a 0.5 mL/min flow rate. The concentration determination was performed at a wavelength of 310 nm for natamycin and a wavelength of 256 nm for voriconazole. The chromatographs were analyzed using the software Empower 3 (Waters^®^). Amounts of 10 µL of samples were injected and the retention time was 0.58 s for natamycin and 1.47 s for voriconazole. Calibration curves were constructed and the R^2^ values obtained were 0.99 for both drugs.

### 2.11. In Vitro Release Studies

The in vitro release study of natamycin and voriconazole from the developed formulations is useful for predicting their in vivo performance. Franz diffusion cells were used to determine the release profile. Visking^®^ dialysis membranes (Medicel^®^ membranes Ltd.) with a 12–14 KDa cut-off (0.784 cm^2^ available surface area) were placed between donor and receptor compartments. An amount of 0.5 mL of formulation was added into the donor compartment, while the receptor compartment was filled in with 6 mL of simulated lacrimal fluid (SLF). The composition of SLF was described in a previous publication by Ceulemans et al. [38]. The Franz cells were kept thermostated at 37 °C and homogenized by magnetic stirring (200 rpm) in a bath during the assay.

The concentration of both drugs was determined by the UPHLC method previously described (see the Ultra-High-Performance Liquid Chromatography section). Apparent permeability (Papp) and flux across the membrane were calculated as described in previous studies [32].

### 2.12. Ex Vivo Corneal Permeability Studies

The ex vivo corneal permeability was carried out using fresh bovine eyes obtained from the local slaughterhouse (Compostelana de Carnes S.L., Santiago de Compostela, Spain). The corneas were excised with a scalpel and immediately placed between the Franz cell donor and the receptor compartment with the outer side towards the donor so that the outer part of the cornea was in contact with the formulation. The receptor compartment was filled in with 6 mL of PBS, while 0.5 mL of formulation was placed in the donor chamber. The Franz cells were kept thermostated at 37 °C and homogenized by magnetic stirring (200 rpm) in a bath during the assay. The concentrations of both drugs were determined by the UHPLC method previously described (see the Ultra-High-Performance Liquid Chromatography section). Apparent permeability (Papp) and flux across the membrane were calculated as described in previous studies [32].

### 2.13. Ocular Irritation Test

BCOP and HET-CAM assays were chosen to evaluate the potential irritation produced on the ocular surface. These methods comply with the 3Rs principles (replacement, reduction, and refinement) as described in Directive 2010/63/EU of the European Parliament and of the Council of 22 September 2010 on the protection of animals used for scientific purposes [39].

#### 2.13.1. Bovine Corneal Opacity and Permeability Assay (BCOP)

##### Corneal Opacity

A variation of the method previously described in the Invittox Protocol nº 437 [40] was carried out to detect potential ocular corrosives and severe irritants using fresh bovine corneas. The assessment of ocular irritation was extensively described in previous studies [32]. Opacity (transmitted light (TL)) and corneal transparency (transmittance values) were measured with a luxmeter (Gossen Mavolux 5032C USB) and a spectrophotometer (Agilent^®^ Cari 60 UV), respectively.

First, the two parameters (opacity and transparency) were measured with fresh corneas. Immediately, the corneas were placed in Franz cells as described in the “Ex vivo corneal permeability studies” section. Then, the corneas were treated for 60 min by introducing 1 mL of PBS into the donor chamber and both parameters were subsequently measured. Following this period, 1 mL of the formulation was added to the corneas and maintained for 10 min, then the formulation was withdrawn, and 1 mL of PBS was added to the corneas and maintained for 120 min. The previous parameters were measured again. Each formulation was tested in triplicate.

##### Corneal Permeability

The corneas used in the “Corneal Opacity” section were placed back into the Franz cells. The receptor chamber was filled in with 6 mL of PBS, while 1 mL of 0.4% (*w*/*v*) fluorescein was placed into the donor chamber. A sample of each Franz cell was collected from the receptor chamber to determine the amount of fluorescein that crossed the treated corneas at 90 min. Fluorescein concentration was measured by a spectrophotometer (Agilent^®^ Cary 60 UV) at a wavelength of 490 nm.

##### Hen’s Egg Test—Chorioallantoic Membrane (HET-CAM)

HET-CAM is described in The Interagency Coordinating Committee on the Validation of Alternative Methods (ICCVAM) [41]. Fertilized broiler chicken eggs were placed in an automatic rotation incubator and kept for 8 days at 38 ± 0.5 °C and 65% relative humidity (RH). At 24 h before the test, the automatic rotation was stopped and on the 9th d of incubation the test was conducted. Each egg was opened, and the inner membrane was removed. Amounts of 0.3 mL of formulation, positive control (0.1% (*w*/*v*) NaOH solution), or negative control (0.9% (*w*/*v*) NaCl solution) were administered onto the surface of the chorioallantoic membrane (CAM). Hemorrhage, vascular lysis, or coagulation reactions were assessed (if applicable) by direct observation of the CAM for 300 s.

### 2.14. Corneal Mucoadhesiveness

The corneal mucoadhesiveness method was designed and described in previous studies [32]. Fresh bovine corneas were excised and fixed to the upper probe of a Universal Testing Machine (Shimadzu^®^ AGS-X Precision Universal Tester). The formulations were introduced into the weighing bottles. The corneas were immersed 2 mm into the formulations for 30 s and then retired to register the force–displacement curve. The bioadhesion work (J) was calculated from the area under the curve (AUC).

### 2.15. PET In Vivo Assay: Quantitative Ocular Permanence Study

The ocular permanence of natamycin/voriconazole on the ocular surface was evaluated in Sprague–Dawley rats by a Positron Emission Tomography (PET) and Computed Tomography (CT) combined system (PET/CT Albira^®^ microPET/CT Bruker Biospin, Woodbridge, CN, USA). The procedure was described in previous studies [32,34,35,42]. All the animal studies and their protocols were approved by the Galician Network Committee for Ethics Research in accordance with the Spanish and EU applicable legislation (86/609/CEE, 2003/65/CE, 2010/63/EU, RD 1201/2005 and RD 53/2013). SNV, AHNV, and LNV were radiolabeled with 2-[^18^F]-fluoro-2-deoxy-D-glucose (^18^F-FDG). An amount of 7.5 µL of formulation containing 0.25 MBq of radioactivity was administrated into each eye of the rat. Immediately, a static PET frame was acquired at 0, 30, 75, 120, 240, and 300 min. Animals were only anesthetized during the image acquisition. Rats were fitted with Elizabethan collars to prevent them from touching their eyes and removing part of the formulation. ROIs (Regions of interest) were manually obtained from the PET images to obtain the ocular remaining formulation (%) curve. Then, data were corrected considering the radioisotope decay (^18^F half-life: 109.7 min). Each formulation was evaluated in quadruplicate. The results were analyzed using a non-compartmental model. The area under the curve (AUC _0_^∞^), terminal half-life (t_1/2_), and mean residence time (MRT) were calculated.

### 2.16. Disc Diffusion Method by the Kirby–Bauer Method

*Candida albicans* ATCC 90231 (C.A 90231), *Candida albicans* ATCC 90028 (C.A 90028), *Paelomyces lilacinus* ATCC 90028 (PL), *Aspergillus fumigatus* (AF), *Paelomyces lilacinus* (PL), and *Fusarium solanii* (FS) were used to perform the diffusion disc method. *Aspergillus fumigatus, Paelomyces lilacinus,* and *Fusarium solanii* isolates were obtained at the bank of the University Clinical Hospital of Santiago de Compostela from FK infections. The isolates were morphologically, biochemically, and molecularly characterized prior to testing. Modified Mueller–Hinton plates were inoculated with suspensions of fungal stock (1 × 10^6^–5 × 10^6^ UFC/mL). The inoculated plates were incubated at 35 °C for 24 h. Then, antifungals discs (containing 20 µL of different formulations (Table 2)) were placed on the inoculated plates. The inhibition zone diameters were measured after incubating the plates containing antifungal discs at 35 °C for 24 h for *Candida Albicans* species and 48 h for *Paelomyces* and *Aspergillus* species.

## 3. Results and Discussion

### 3.1. Phase Solubility Diagrams

Due to the low water solubility of natamycin (30–50 mg/L [43]), improving its aqueous solubility was essential for the development of new ophthalmic formulations.

Solubility diagrams were created with HPβCD and HPγCD (Figure 1). The cyclodextrins were chosen based on their large size (1135 and 1761 g/mol, respectively) and the volume of their cavities. In addition, both have been previously studied for the development of ophthalmic formulations, demonstrating their compatibility with this route [44,45]. In addition, the literature prior to these studies has demonstrated the ability of cyclodextrins to reduce ocular irritation, improve corneal permeability, and increase the bioavailability of drugs with very low water solubility [46,47].

The phase solubility diagrams for natamycin/cyclodextrin inclusion complexes (Figure 1) were A_N_ type. These data agree with the study of Koontz and Marcy [43]. They evaluated the solubility of natamycin with three natural cyclodextrins (αCD, βCD, and γCD) and with HPβCD, a βCD derivative. However, A_N_-type phase solubility diagrams occurred only with γCD and HPβCD. In the solubility studies, an initial solubility of 0.0353 ± 0.014 mg/mL was obtained for natamycin without cyclodextrin. Similar values were obtained in Koontz and Marcy’s study (0.034 mg/mL) [43].

The A_N_-type phase solubility diagram shows the self-association of cyclodextrin aggregates or their complexes, which can decrease the drug solubility [48].

The possible existence of cyclodextrin aggregates or their complexes was evaluated for saturated solutions of natamycin in 40% (*w*/*v*) HPβCD solution (Figure 2b) and saturated solution of natamycin in 40% (*w*/*v*) HPβCD solution and 1% (*w*/*v*) voriconazole solution (Figure 2a). The resulting Transmission Electron Microscopy (TEM) images (Figure 2) showed the formation of nanometric spheric aggregates. The saturated solution of natamycin in 40% (*w*/*v*) HPβCD solution showed an average particle size of 80.25 ± 35.81 nm. However, the saturated solution of natamycin in 40% (*w*/*v*) HPβCD solution in the presence of 1% (*w*/*v*) voriconazole showed larger particles with an average particle size of 148.96 ± 32.89 nm.

The cyclodextrin aggregation occurs because of intermolecular hydrogen bonds among cyclodextrin hydroxyl groups (OH). These hydrogen bonds lead to the assembly of the dissolved CD molecules into CD aggregates. The resultant nanoparticles, formed by drug/CD complexes, have demonstrated the ability to improve drug permeation across corneal membranes better than the individual inclusion complexes. The CD aggregates can behave in the same way as nanosystems, controlling drug release and increasing residence time at the site of administration [49]. Loftsson and Stefansson (2007) described a system based on dexamethasone/γCD complex aggregates intended for topical ocular administration with sustained delivery and enhanced bioavailability of dexamethasone [50]. Other authors have described similar results for drugs like dorzolamide [51], irbesartan [52], or cyclosporin A [53], among others.

Table 3 shows the apparent stability constant (K_1:1_), complexation efficiency (CE), natamycin:cyclodextrin complex molar ratio (D:CD), coefficient of determination (R^2^), and water natamycin solubility S_0._

K_1:1_ was calculated on the linear portion of the phase diagram due to the negative deviation previously shown [54]. The HPβCD showed a higher K_1:1_ than HPγCD, suggesting that the natamycin/HPβCD interactions were stronger than those of natamycin/HPγCD. Furthermore, HPβCD obtained the best solubilization properties for the natamycin and showed higher CE values than HPγCD (0.061 and 0.049, respectively). D:CD values were high for both cyclodextrins (1:17.50 and 1:21.41, respectively), but HPβCD showed the lowest value, so its bioavailability would be better than that of HPγCD [55].

### 3.2. Natamycin Solubility with HPβCD and Different Hydrophilic Polymers

The drug solubility in the presence of hydrophilic polymers can be enhanced by ternary complexation [56]. Natamycin solubility with 20% (*w*/*v*) HPβCD and different hydrophilic polymers is shown in Table 4. Data were compared with natamycin solubility in the absence of polymers. These data do not show significant differences in natamycin solubility when the hydrophilic polymers were added to the system.

### 3.3. Natamycin and Voriconazole Solubility with HPβCD and Voriconazole

Figure 3a shows the natamycin concentration reached with 20%, 30%, and 40% (*w*/*v*) of HPβCD and 1% (*w*/*v*) of voriconazole. The voriconazole concentration remained stable in all HPβCD solutions. Natamycin concentration increased from 6.175 ± 0.658 to 7.951 ± 0.389 mg/mL with increasing HPβCD concentration. These data agree with the data obtained by interpolation in the phase diagram (Table 5). Figure 3b shows the voriconazole concentration reached with 20%, 30%, and 40% (*w*/*v*) of HPβCD and 0.4% (*w*/*v*) of natamycin. Voriconazole concentrations obtained in the presence of 0.4% (*w*/*v*) natamycin and 20%, 30%, and 40% (*w*/*v*) of HPβCD were 12.43 ± 4.71, 15.937 ± 5.05, and 25.04 ± 1.44 mg/mL, respectively. The concentrations of voriconazole obtained in the presence of natamycin are similar to those obtained without natamycin in previous studies [32] (see Table 5).

These results suggest that the incorporation of natamycin into the voriconazole/HPβCD complexes solution does not affect the solubility of voriconazole in the presence of cyclodextrin. To clarify this statement, a competition study of the two drugs for cyclodextrin was performed using Nuclear Magnetic Resonance (NMR).

### 3.4. Nuclear Magnetic Resonance (NMR) Studies

Molecular interactions are essential for many biological processes. The binding process is promoted by the establishment of a number of favorable non-covalent forces between the molecules that interact and there is a dynamic equilibrium between association and dissociation events. NMR is one of the methods for the screening of ligands that bind to a receptor and detect the ligand binding epitope and/or receptor binding site with quantitative results [57,58].

#### 3.4.1. Detection of Binding Interaction between Natamycin and HPβCD

The ^1^H-NMR signal assignment of natamycin in CD_3_OD was obtained from the ^1^H and ^13^C predicted spectrum at 800 MHz in the Human Metabolomic Database [59,60] in concordance with the experimental 2D edited-HSQC spectrum obtained by us in Appendix A. The comparison of the ^1^H spectrum of pure natamycin in CD_3_OD and the mixture natamycin:HPβCD 1:1 in D_2_O denotes relevant changes in the chemical shifts (i.e., CSPs) that could be due to either a change in the conformation due to the solvent and/or its binding to HPβCD. To further investigate the possibility of binding interaction between ligand natamycin and receptor HPβCD in water solution, it was tested by STD experiments [30,31]. The STD^off^ reference ^1^H spectrum of the mixture natamycin:HPβCD 1:1 is shown in Figure 4aa. STD^on-off^ spectra were measured to determine possible intermolecular contacts between the ligand and receptor in the mixture. A requisite for the STD^off-on^ experiment is that the on-saturation should only affect the signal/s of one of the two components in the mixture. In this case, placing the on-saturation in the region where the majority of the protons of HPβCD appear in the ^1^H spectrum, between 3.2 and 4.6 ppm, should be avoided because in this region a few signals of natamycin are also present (Appendix A). For this reason, the on-saturation was placed over specific aromatic signal/s of the ligand natamycin that are well isolated in the ^1^H spectrum. The STD^on-off^ spectra (Figure 4aa–ad) show the STD responses; some of them are intramolecular NOE contacts in natamycin while those that appear well extended and with broad features in the region between 3.2 and 4.6 ppm were assigned to HPβCD. This result confirms that there is binding affinity between natamycin and HPβCD.

#### 3.4.2. NMR Titration Competition Study of Natamycin and Voriconazole for Binding to HPβCD

Having established the affinity between natamycin and HPβCD, the binding affinity of natamycin was tested in a competition experiment with the ligand voriconazole in water. The affinity of this later ligand for HPβCD was tested in our laboratory and showed the formation of an inclusion complex of stoichiometry 1:1 with a dissociation constant K_D_ of 250 mM [32].

Under conditions of weak binding equilibrium of a ligand to a receptor (typically for K_D_ in the 1 to 1000 mM range), a chemical shift titration is a feasible method to map a ligand-binding site on a target receptor such as a protein [58] or a cyclodextrin [61] and may serve to estimate the K_D_ of the equilibrium. The basis of this method under weak binding is that there is a fast exchange equilibrium between the free and bound species in the NMR time scale, and the observed chemical shift d^obs^ is a weighted average given by Equation (1) [57]:d^obs^ = c^free^ ⋅ d^free^ + c ^bound^ ⋅ d^bound^
(1)
where d^free^ and d^bound^ are the chemical shifts in the free and bound states, respectively, and c^free^ and c^bound^ are the molar fraction of the species in the free and bound states, respectively, with c^free^ + c^bound^ = 1. Chemical Shift Perturbations (CSPs) can be quantified as d^obs −^ d^free^ (in units of Hz) for any signal in the spectrum at any point in the titration to map the ligand-binding site.

The ^1^H-NMR spectra of the titration competition assay for the mixtures prepared of natamycin, voriconazole, and HPβCD are shown in Figure 4b. The experiment was carried out by the addition of natamycin to a sample containing equimolar concentrations of voriconazole and HPβCD. It can be seen in Figure 4b that in the course of the titration certain signals of voriconazole have CSPs, while all the signals of natamycin remain at the same chemical shift and only increase their intensity. Qualitatively, this observation strongly suggests that natamycin competes for the same binding site of HPβCD and has a stronger affinity than voriconazole because as the concentration of natamycin is raised in the titration the signals of voriconazole move towards their characteristic values in the free state d^free^, which can be explained by a higher molar fraction c^free^ in Equation (1). In Appendix A are given the CSPs of several signals of voriconazole in this competition assay. The data was fit to a competitive binding model (described in Appendix A) and provided a K_I_ = 0.334 mM, which is a value that represents a ca. 700 times higher affinity of natamycin than voriconazole for binding to HPβCD. In our previous work, the molecular model of the complex HPβCD:voriconazole [32] showed that voriconazole can be inserted almost completely into the cavity of HPβCD. Natamycin is a larger molecule than voriconazole and can only be incorporated partially inside the cavity of HPβCD as can be seen in the optimized molecular mechanics model of Figure 5. In this model, the double bonds of natamycin are disposed towards the most hydrophobic side of HPβCD in proximity to the hydroxypropyl pendant chains and the two polar six-member rings of natamycin disposed towards the most hydrophilic side of HPβCD.

### 3.5. Transparency

One of the problems that usually leads to a discontinuation of the FK treatment is the number of required instillations (1 drop every 1–2 h). In addition, some eye drops may cause blurred vision due to their organoleptic or physicochemical characteristics (e.g., color, high viscosity, among other factors), increasing the treatment dropout. Natacyn^®^ is a cloudy white-to-yellow aqueous suspension containing natamycin particles that can cause blurred vision after administration. For this reason, transparency measurements were carried out.

Natacyn^®^ showed transmittance values close to 0 in all light ranges. The results obtained from transparency measurements are shown in Figure 6. All formulations (SLV, AHNL, and LNV) were practically transparent (transmittance ≃ 100%) in the infrared (780 nm onwards) and visible light range (from 380 to 780 nm). The decrease in transmittance values in the UV range for all the formulations was possibly associated with the presence of molecules that absorb in the UV range, such as natamycin 310 nm or voriconazole 256 nm.

### 3.6. Osmolality and pH Measurements

Table 6 shows the pH, osmolality, and viscosity data of the ocular formulations. The viscosity of AHNV (265.5 ± 37.56 mPa·s) is higher than the viscosity obtained for LNV (54.292 ± 2.88). The viscosity data for Liquifilm^®^ were 5.827 ± 0.284 mPa·s.

The osmolality data obtained for SNV and AHNV were lower than the osmolality data for Liquifilm^®^ formulation (LNV). The high osmolality value of LNV (500 ± 3.46 mOsm/kg) is due to the osmolality values of the Liquifilm^®^ vehicle (256.5 ± 8 mOsm/kg [62]) used to formulate the LNV formulation. The osmolality values of all formulations are higher than the physiological value (290 mOsm/kg); however, in an in vivo system, the high precorneal clearance would protect the ocular surface from hyperosmolality by removing the formulation from the corneal surface [63,64].

The new formulations (SNV, AHNV, and LNV) showed pH values within the tolerable range of the eye (pH range 4 to 8) [46].

### 3.7. In Vitro Release Studies

The in vitro release profile of the resulting formulations is shown in Figure 7. Papp (cm/s), flux (µg/min), and R^2^ are shown in Table 7. All profiles were fitted to a Korsmeyer–Peppas model, and the R^2^ values were >0.96 for all tested formulations. The resulting *n* values show that both drugs are released by a Fickian diffusion process (*n* ≤ 0.45) from all the formulations (SNV, AHNV, and LNV), while natamycin in the Natacyn^®^ formulation presents an anomalous diffusion mechanism (0.45 < *n* > 0.89) [65]. However, this value of *n* for Natacyn^®^ (0.629) is due to the compensation of the amount of natamycin that diffuses with the amount of natamycin that dissolves from the solid particles of the suspension in the diffusion process.

With respect to Natacyn^®^, the natamycin particles must be dissolved in the medium to diffuse through the dialysis membrane, unlike the newly developed formulations in which the natamycin was already dissolved. For this reason, Natacyn^®^ Papp values (0.05·10^−6^ cm/s) were lower than SNV, HAV, and LNV release values (1.398·10^−6^, 1.010·10^−6^, and 1.810·10^−6^ cm/s, respectively).

### 3.8. Ex Vivo Corneal Permeability Studies

The fungal infection, which usually occurs on the corneal surface, reaches the internal ocular structures such as the aqueous or vitreous humor, causing endophthalmitis. The corneal epithelium consists of a cell layer bound by tight junctions that resist the permeability of large drug molecules such as natamycin (molecular weight of 665.75 g/mol), preventing them from reaching the internal structures of the eye. For this reason, it is common in clinical practice to carry out corneal scrapings to remove the epithelium and, thus, favor the penetration of drugs. Overall, the infections usually lead to the epithelium breakdown [66]. The corneal permeability of the developed formulations (SNV, HAV, and LNV) and Natacyn^®^ was studied to know the natamycin and voriconazole capacity to go through the corneal structure with and without corneal epithelium.

The corneal permeation data for the epithelized and de-epithelized bovine corneas are presented in Figure 8. Apparent permeability (Papp), flux, and lag time data were calculated for both studies and are represented in Figure 9.

Natamycin and voriconazole Papp values were higher in absence of corneal epithelium than in the presence of corneal epithelium (see data details in Figure 9a,b). Natacyn^®^ did not show natamycin permeability in the presence of corneal epithelium but it was improved in de-epithelized corneas (7.17·10^−9^ ± 8.90·10^−9^ cm^2^/s). Other authors, such as O’Day et al. [67], also described an improvement in the passage of natamycin when the corneal epithelium was removed. 

The administration of natamycin solubilized with HPβCD (In SNV, AHNV and LNV formulations) improved the passage of natamycin in the presence of corneal epithelium. This may be because cyclodextrins decrease the resistance of the aqueous layer exerted by the tear and the ocular mucosa. Moreover, cyclodextrins enhance the passage across the cornea by the extraction of cholesterol from the corneal epithelium [68]. Lorenzo Veiga et al. [69] developed a natamycin micelle formulation in which the values of the cumulative amount of permeated natamycin were below 0.01 µg/cm^2^ at 5 h of permeation in epithelized corneas. The quantities of natamycin permeated with SNV, AHNV, and LNV were 0.15 ± 0.06, 0.37 ± 0.20, and 0.67 ± 0.66 µg/cm^2^, respectively at 5 h of permeation in epithelized corneas.

The values obtained for voriconazole showed, both in the presence and absence of epithelium, better Papp and flux values than natamycin in the presence of corneal epithelium (Figure 9a–d).

In addition, in the presence of epithelium, the lag time values show that voriconazole takes less time to cross the cornea than natamycin (see data details in Figure 9e,f). However, in the absence of epithelium, the time lag data (Figure 9e) decreased considerably for natamycin (SNV 36.77 ± 3.028, AHNV 37.85 ± 11.22, and LNV 39.87 ± 6.144 min) with an average of 38.49 ± 1.19 min instead of 102.98 ± 5.81 min with corneal epithelium.

Permeability, flux, and lag time data show that the limiting step in the penetration of natamycin through the cornea is mainly the passage through the corneal epithelium.

### 3.9. Ocular Irritation Test

#### 3.9.1. Bovine Corneal Opacity and Permeability Assay (BCOP)

BCOP data (Figure 10) showed that there were no significant modifications in transparency and opacity after treating the bovine corneas with SNV, AHNV, and LNV for 10 min. An ANOVA test showed no significant differences between C- (PBS) and all formulations tested.

The data obtained in the corneal permeability test showed no passage of fluorescein and therefore the tested formulations did not modify corneal permeability, maintaining the integrity of the cornea.

#### 3.9.2. Hen’s Egg Test—Chorioallantoic Membrane (HET-CAM)

Natamycin/voriconazole formulations (SNV, AHNV, and LNV) were tested on the egg’s chorioallantoic membrane (Figure 11). None of the formulations showed vessel modifications (hemorrhage, lysis, or coagulation at 5 min) compared to a NaOH solution (positive control). Consequently, all formulations can be considered non-ocular irritants.

### 3.10. Corneal Mucoadhesiveness

The study of the mucoadhesive properties of the topical ophthalmic formulations can predict the prolonged permanence on the ocular surface improving the effectiveness of the treatment.

The bioadhesion work for all formulations is represented in Appendix A. All formulations showed similar values of bioadhesion work (mJ) (SNV: 0.0251 ± 0.002 mJ, AHNV: 0.0242 ± 0.005 mJ, LNV: 0.0249 ± 0.002 mJ). These results can be explained because, during the traction stage of the assay, the bond between the cornea and the formulation is broken within the formulation itself, not at the corneal surface. As all the formulations are low-viscosity systems, the interaction between the vehicle molecules is low, and therefore lower than the bioadhesive forces between the cornea and the formulation itself. A one-way ANOVA test was performed, and no significant differences were found in the bioadhesion work data for SNV, AHNV, and LNV. For this reason, an in vivo permanence study was carried out by PET to quantify the amount of formulation that remains on the ocular surface over time.

### 3.11. PET In Vivo Assay: Quantitative Ocular Permanence Study

An eye drop formulation with low ocular permanence cannot ensure the necessary time for the drug to diffuse through the corneal tissue. A PET in vivo assay was carried out to quantify the amount of formulation remaining on the ocular surface over time.

The semi-logarithmic plot (Figure 12) shows the data of the clearance rate of the formulations as a function of the remaining radioactivity in the eye. Table 8 shows the pharmacokinetic parameters (elimination constant *K*, *t_1/2_*, AUC _0_^∞^, and MRT) obtained by fitting the formulation percentage remaining in the eye over time.

One-way ANOVA and Kruskal–Wallis tests were performed, and no significant differences were found for any formulation pharmacokinetic parameters. This suggests that HPβCD is responsible for ocular permanence independently of the added polymer. Previous studies obtained similar results, proving the mucoadhesive capacity of HPβCD at high concentrations (40% (*p*/*v*)) [35].

The results show that there is a high clearance in the first few minutes due to the loss of the formulation by blinking (Figure 13). This is consistent with the mucoadhesion values indicating that the breakage of the bioadhesive bond occurs at the level of the vehicle itself. After those first few minutes, the layer of formulation that remains bioadhered to the cornea is slowly removed.

Although the addition of polymers such as HA in AHNV and PVA in LNV has not increased the ocular permanence compared with the solution without polymers (SNV), the advantages of the presence of polymers must be considered. The addition of high molecular weight HA (>1000 KDa) [70] to eye drops can promote faster corneal wound healing. This is due to the binding of the HA to the CD44 protein (CD44 receptor is expressed when there is damage in the corneal epithelium), which enhances the migration and regeneration of epithelial cells [71]. Furthermore, the expression of inflammatory cytokines (e.g., IL-1beta and MMP-9) is decreased following the HA administration, suppressing inflammatory responses [72,73]. In addition, high molecular weight HA retains water, increasing tear film stability and decreasing the friction during the blink. In addition, HA increases eye hydration [74]. However, PVA is used in ophthalmic formulations as a lubricant and to improve ocular surface hydration, which can increase the sense of well-being during the treatment of FK.

### 3.12. Disc Diffusion Method by the Kirby-Bauer Method

Inhibition zone diameters (Appendix A) obtained in the Kirby–Bauer Disc Diffusion Method are shown in Table 9. The inhibitory zone is influenced by several parameters, including the culture medium, the drug diffusion capacity, the amount of inoculum, the time of microorganism generation, the sensitivity to the antifungal, or the incubation period [75]. SN shows larger inhibitory zone diameters for all fungal species than NTC. These results are due to the enhanced diffusion of natamycin from the disc into the inoculum when natamycin is complexed with HPβCD. SN and NTC show no inhibitory activity for PL. PL resistance to natamycin has been previously reported in cases of FK [76,77].

The inhibition zone diameters obtained for the formulation with both drugs solubilized with HPβCD (SNV) show slightly higher values than those obtained with the single drug formulations (SN and SV) for all species except for AF and FS.

The combined formulation of natamycin and voriconazole was effective for the species tested. However, to confirm an additive or synergistic effect produced by the combination of natamycin and voriconazole, further in vitro studies using other more representative techniques such as the checkerboard method [78] or ETEST^®^ strips [79] would be necessary. Moreover, additional in vivo and ex vivo studies would be required to assess the antifungal efficacy.

## 4. Conclusions

Natamycin and voriconazole formulations were developed as a new alternative for the treatment of FK. The formulations were developed to have characteristics suitable for topical ophthalmic administration.

Solubility and NMR studies demonstrated the formation of stable complexes between natamycin and HPβCD where the double bonds of natamycin are arranged towards the more hydrophobic side of HPβCD and the polar rings of natamycin are arranged on the more hydrophilic side of HPβCD. Furthermore, NMR results showed that natamycin competes with voriconazole for the same binding site of HPβCD although this did not affect the solubility of voriconazole in the formulations due to the presence of free cyclodextrin molecules.

In addition, the formation of aggregates of HPβCD molecules and their complexes with natamycin and voriconazole was observed by TEM. These can control drug release, improve residence time, and enhance their permeability across the cornea.

The pH, osmolality, viscosity, and transparency values were found to be within the accepted range for ophthalmic topical formulations. In vitro release studies were successfully carried out and Fickian-type diffusions were obtained for all formulations developed. All formulations showed an improvement in transcorneal permeability in the presence or absence of corneal epithelium compared to Natacyn^®^. In addition, the ocular toxicity studies performed (BCOP and HET-CAM) showed that the formulations are safe. Ex vivo and in vivo mucoadhesion studies suggested that the mucoadhesive capacity of the formulations is due to the presence of HPβCD and is not increased by the addition of HA or PVA. Antifungal activity studies demonstrated the ability to inhibit the growth of several fungal species. All these results concluded that the formulations developed in the present study could significantly improve the treatment of FK. Additionally, formulations can also be prepared using only one of the drugs, making it a versatile pharmaceutical system that can be tailored to meet the different needs of patients. Therefore, they could be used as a first-choice treatment in cases of FK where the causative agent is unknown, species resistant to one of the antifungal agents are suspected, or no commercial drug is available.

## Figures and Tables

**Figure 1 pharmaceutics-15-00035-f001:**
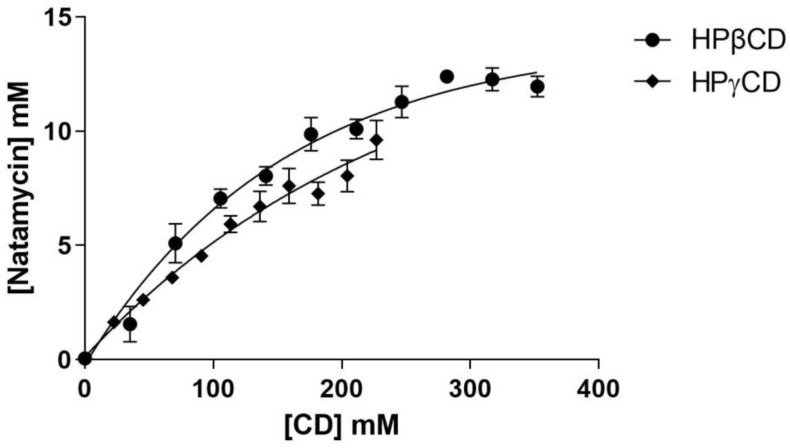
Phase solubility diagrams for natamycin, obtained with 2 different types of cyclodextrin (HPβCD and HPγCD) at 25 °C in water (mean ± SD, *n* = 6).

**Figure 2 pharmaceutics-15-00035-f002:**
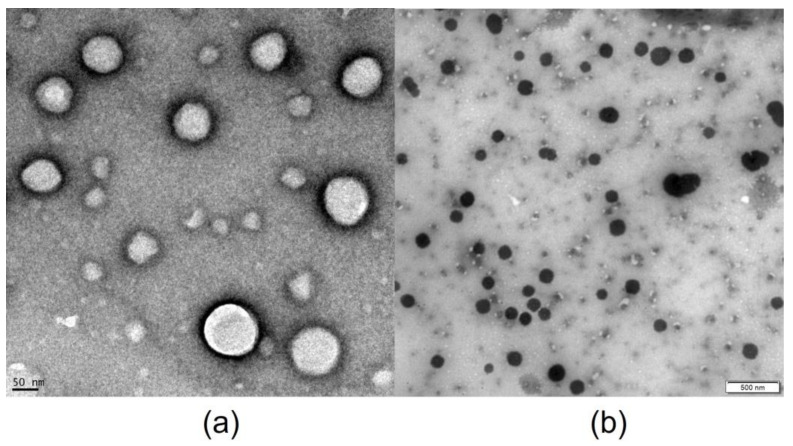
Transmission Electron Microscopy (TEM) images of saturated solution of (**a**) natamycin in 40% (*w*/*v*) HPβCD solution and 1% (*w*/*v*) voriconazole solution and (**b**) saturated solution of natamycin in 40% (*w*/*v*) HPβCD solution.

**Figure 3 pharmaceutics-15-00035-f003:**
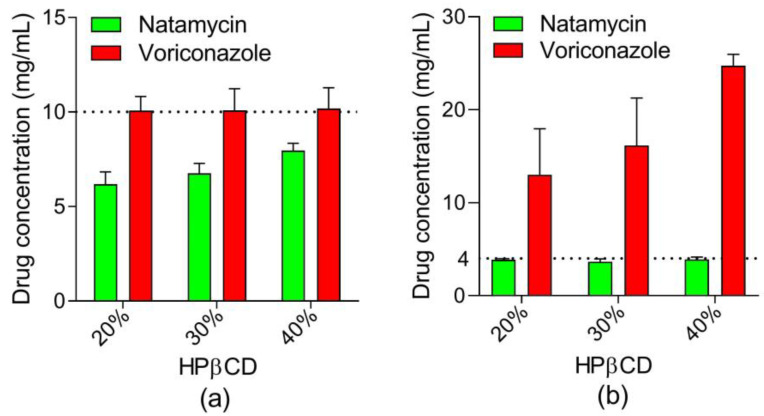
(**a**) Voriconazole concentration obtained in presence of 4 mg/mL of natamycin and different concentrations of HPβCD. (**b**) Natamycin concentration obtained in presence of 10 mg/mL of voriconazole and different concentrations of HPβCD.

**Figure 4 pharmaceutics-15-00035-f004:**
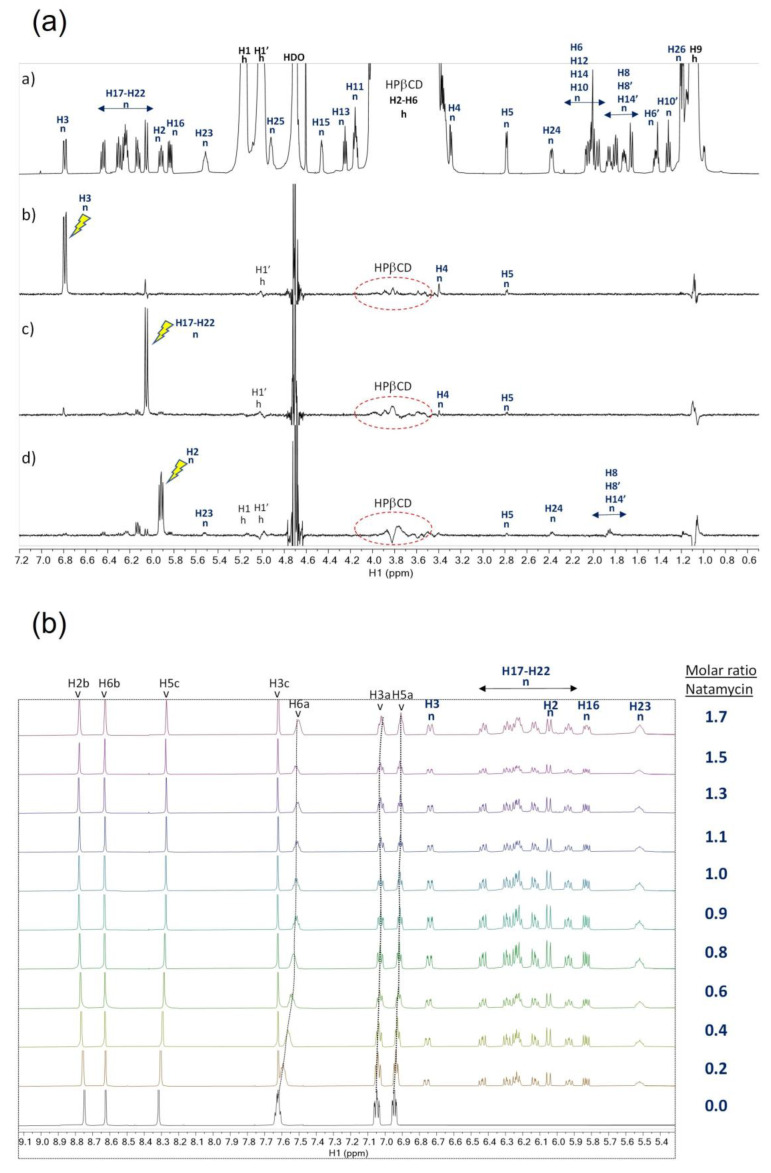
(**a**) NMR spectra of natamycin:HPβCD 1:1 in D_2_O showing the assignment of signals of natamycin (n) and HPβCD (h). (a) ^1^H reference spectrum. (b) STD^off-on^ spectrum with on-saturation at 6.79 ppm (H-3 signal of n). (c) STD^off-on^ with on-saturation at 6.05 ppm (H-17 to H-22 signal of n). (d) STD^off-on^ with on-saturation at 5.91 ppm (H-2 signal of n). The atom numbering used to identify the signals of voriconazole follows S1. (**b**) NMR titration competition assay with natamycin at a constant molar ratio voriconazole:HPβCD 1:1. Stack of spectra showing the aromatic region of the ^1^H-NMR spectrum during the titration. The atom numbering used to identify voriconazole (v) and natamycin (n) follows S1. Stripped lines were drawn to guide the eye for the changes in chemical shift (i.e., CSPs) of certain signals of voriconazole.

**Figure 5 pharmaceutics-15-00035-f005:**
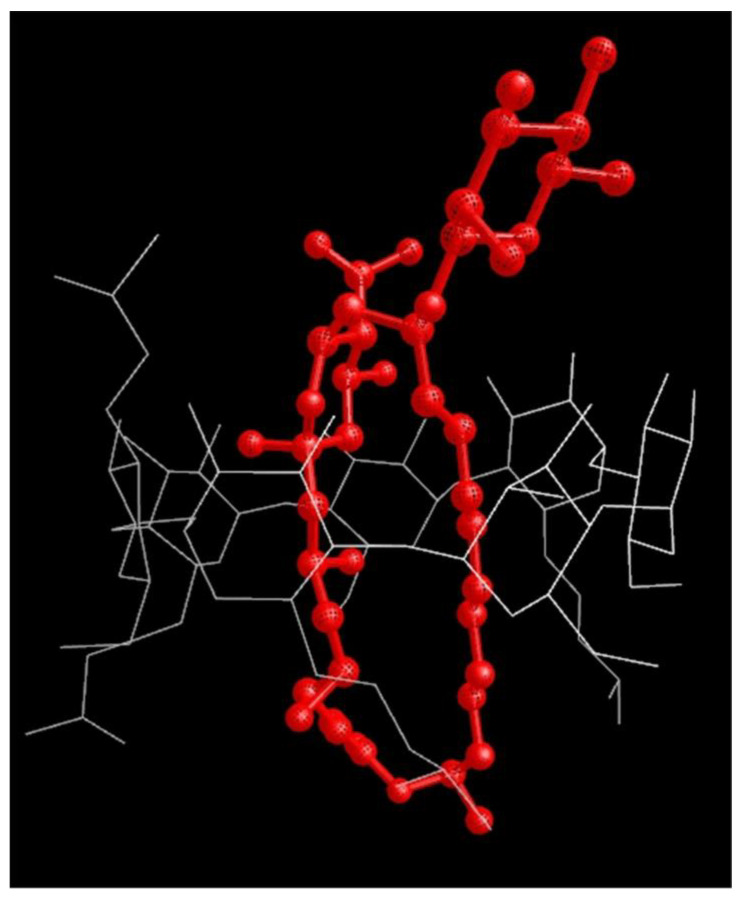
Optimized molecular mechanics model of the natamycin complex with HPβCD. Red lines represent natamycin and white lines represent HPβCD.

**Figure 6 pharmaceutics-15-00035-f006:**
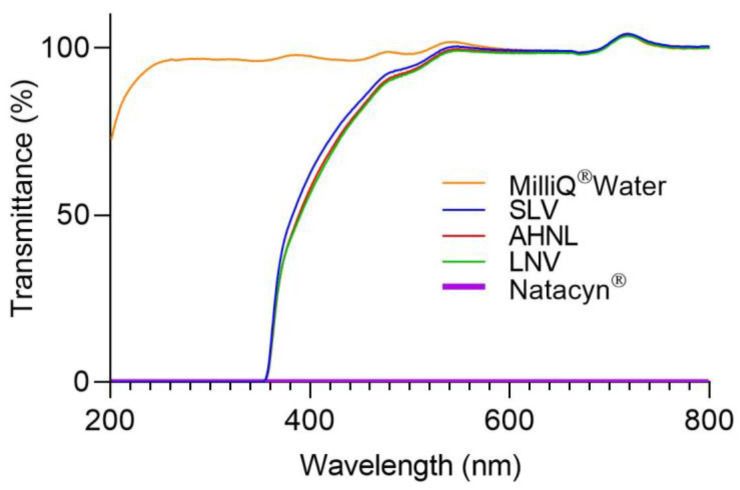
Scan light transmittance from 200 to 800 nm for the formulations including natamycin. MilliQ^®^ water was used as a transparent formulation in all light ranges.

**Figure 7 pharmaceutics-15-00035-f007:**
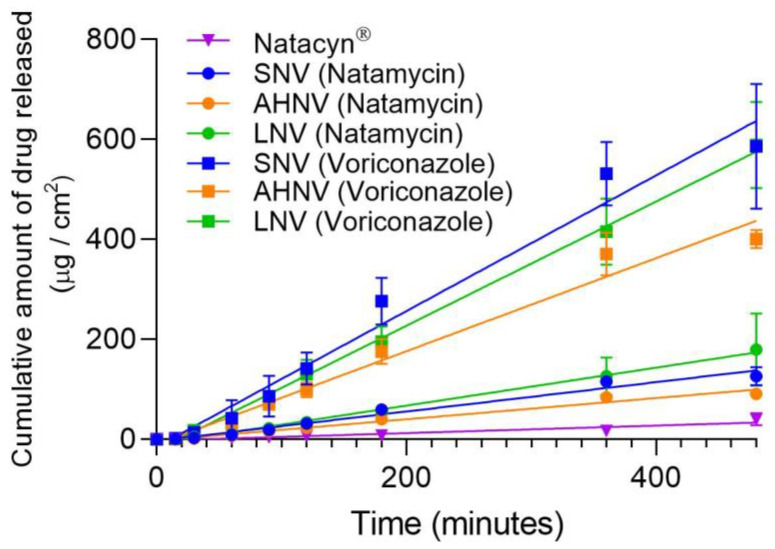
In vitro release profiles of natamycin/voriconazole formulations. All data were fitted to zero-order kinetics.

**Figure 8 pharmaceutics-15-00035-f008:**
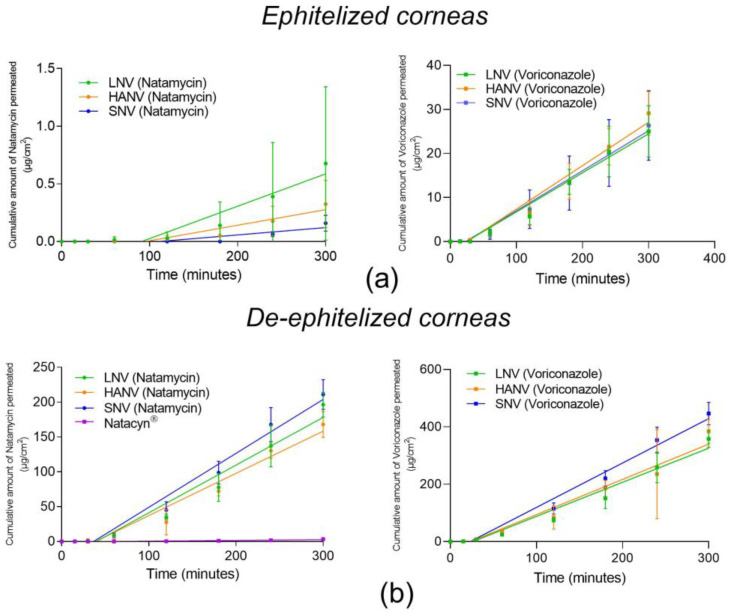
Cumulative amount of drug permeated (µg/cm^2^) through epithelized (**a**) bovine and de-epithelized (**b**) corneas for natamycin (left) and voriconazole (right).

**Figure 9 pharmaceutics-15-00035-f009:**
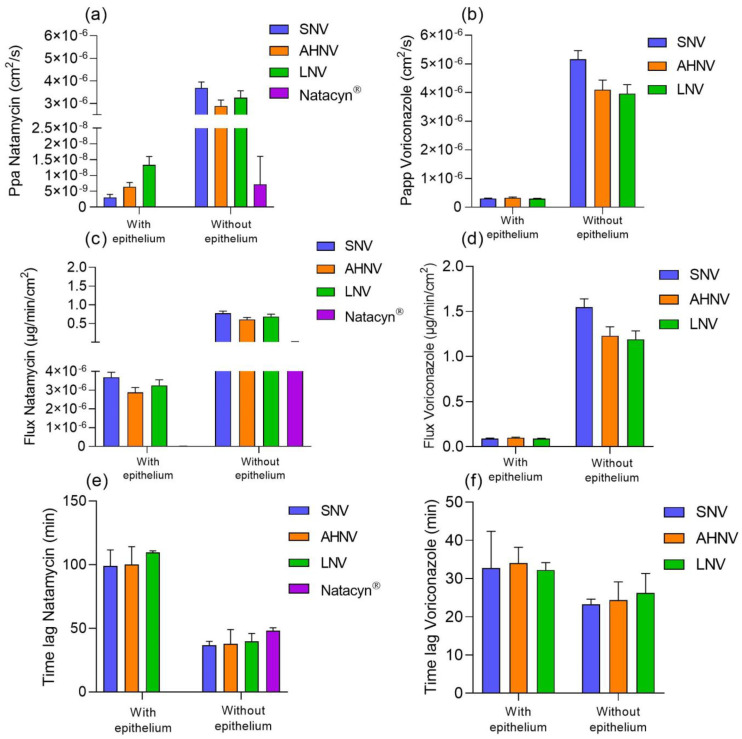
(**a**) Apparent permeability (Papp) of natamycin, (**b**) apparent permeability (Papp) of voriconazole, (**c**) flux of natamycin, (**d**) flux of voriconazole, (**e**) lag time of natamycin, and (**f**) lag time of voriconazole of natamycin and voriconazole formulations across epithelized and deepithelized bovine corneas. * Natacyn^®^ permeability values are not shown because there was no permeability of natamycin from this formulation.

**Figure 10 pharmaceutics-15-00035-f010:**
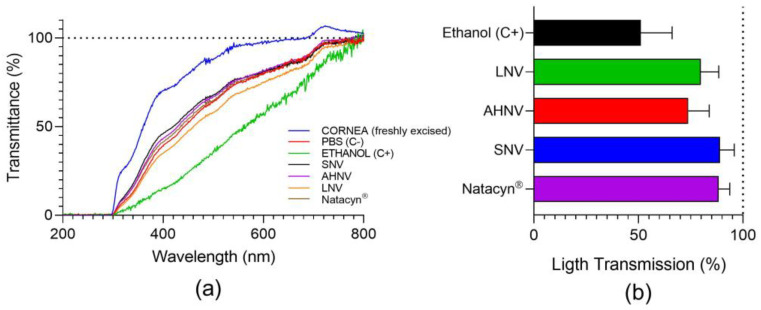
(**a**) Transmittance values in the ultra-visible light spectrum (200–800 nm) of bovine corneas treated 10 min with SNV, AHNV, and LNV. Values are compared with ethanol (C+: positive control), PBS (C−: negative control), and untreated corneas. (**b**) Transmitted light (%) (opacity) values of bovine corneas treated with SNV, AHNV, and LNV. Data were compared with ethanol (C+: positive control).

**Figure 11 pharmaceutics-15-00035-f011:**
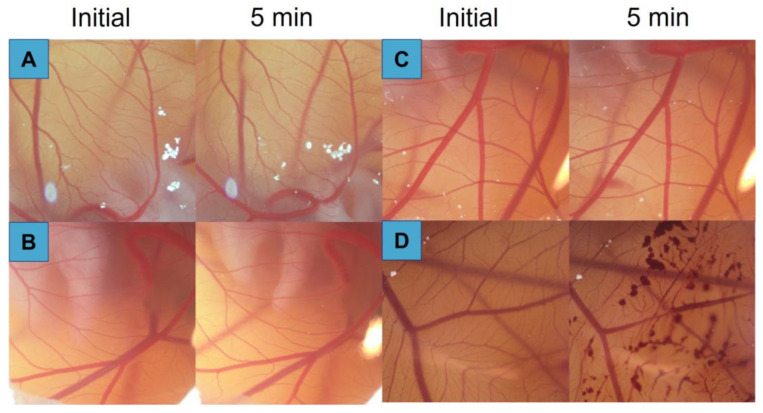
5 min post-instillation images of Hen´s egg test on the chorioallantoic membrane (HET-CAM) for different natamycin/voriconazole formulations. (**A**) SNV; (**B**) AHNV; (**C**) LNV; (**D**) NaOH 0.1 M.

**Figure 12 pharmaceutics-15-00035-f012:**
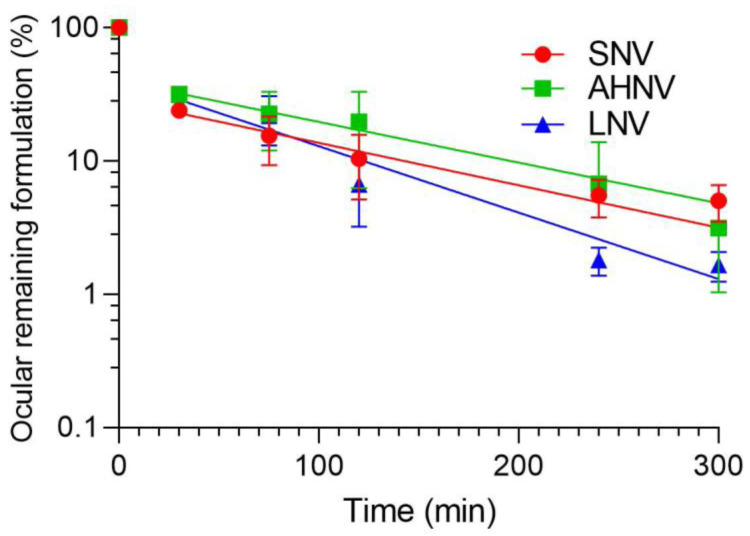
SNV, AHNV, and LNV clearance ratio from the ocular surface determination by PET. Ratio CT/C_initial_ was calculated assuming C_initial_ value obtained in the Regions of Interest (ROI).

**Figure 13 pharmaceutics-15-00035-f013:**
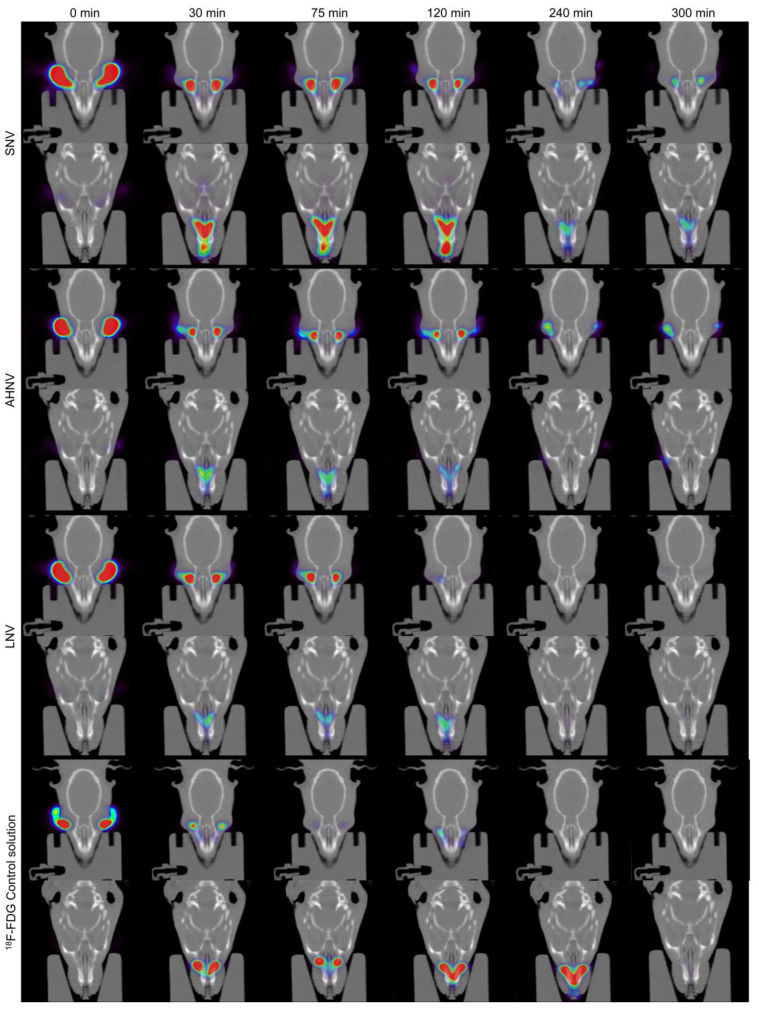
Coronal PET/CT images of rat eyes treated with SNV, AHNV, and LNV over time. Data were compared with a ^18^F-FDG control solution. The amount of formulation on the ocular surface is coded on a color scale: blue areas show a low radioactive activity; red areas show high radioactive activity.

**Table 1 pharmaceutics-15-00035-t001:** Composition of polymeric solutions to assess the potential increase in water solubilization efficiency of natamycin.

Solution	HPβCD % (*w*/*v*)	Polymer % (*w*/*v*)
I	20	-
II	20	0.5% PVA
III	20	1% PVA
IV	20	0.4% HA
V	20	0.1% P407
VI	20	0.5% MC

**Table 2 pharmaceutics-15-00035-t002:** Composition of tested formulations in Kirby–Bauer Disc Diffusion Method.

Formulation	Composition
SV	40% (*w*/*v*) HPβCD + 1% (*w*/*v*) voriconazole
SN	40% HPβCD + 0.7% (*w*/*v*) natamycin
SNV	40% HPβCD + 0.7% (*w*/*v*) natamycin + 1% (*w*/*v*) voriconazole
VFEND	Vfend^®^ (1% (*w*/*v*) voriconazole + 16% (*w*/*v*) SBEβCD)
NTC	Natacyn^®^ (5% (*w*/*v*) natamycin)

**Table 3 pharmaceutics-15-00035-t003:** Values for K1:1, CE, and the D:CD ratio, obtained from the natamycin/cyclodextrin complex in water at 25 °C.

Inclusion Complex	R^2h^	K_1:1_ (M^−1^) *	CE (M)	S_0_ (M)	D:CD (mol:mol)
Natamycin/HPβCD	0.9717	1102.32 ± 89.09	0.061	5.50·10^−5^ ± 2.06·10^−5^	1:17.50
Natamycin/HPγCD	0.9943	891.08 ± 26.39	0.049	5.50·10^−5^ ± 2.06·10^−5^	1:21.41

* K_1:1_ calculated using S_0_ (free drug solubility).

**Table 4 pharmaceutics-15-00035-t004:** Natamycin concentration in presence of 20% (*w*/*v*) HPβCD and different hydrophilic polymers.

HPβCD and Polymers Solutions	Natamycin Concentration (mg/mL)
20% HPβCD	5.151 ± 0.206
20% HPβCD + 0.5% PVA	5.740 ± 0.867
20% HPβCD + 1% PVA	5.117 ± 0.484
20% HPβCD + 0.4% AH	4.625 ± 0.464
20% HPβCD + 0.1% P407	4.222 ± 0.574
20% HPβCD + 0.5% MC	4.181 ± 0.309

**Table 5 pharmaceutics-15-00035-t005:** Concentration of natamycin and voriconazole in HPβCD solutions obtained from interpolation in the phase diagrams (Figure 1). The HPβCD phase diagram of voriconazole was published in previous studies [32].

	Natamycin (mg/mL)	Voriconazole (mg/mL)
20% HPβCD (*w*/*v*)	6.351	15.203
30% HPβCD (*w*/*v*)	7.639	22.252
40% HPβCD (*w*/*v*)	8.372	29.301

**Table 6 pharmaceutics-15-00035-t006:** pH, osmolality, and viscosity results.

Formulation	pH	Osmolality (mOsm/kg)	Viscosity (mPa·s)
SNV	6.10 ± 0.16	304 ± 3.46	9.853 ± 0.326
AHNV	6.34 ± 0.08	344 ± 3.46	265.5 ± 37.56
LNV	7.09 ± 0.02	500 ± 3.46	54.29 ± 2.880

**Table 7 pharmaceutics-15-00035-t007:** In vitro release data from natamycin/voriconazole formulations.

Formulation	Papp (cm/s)	SE⋯10^−7^	Flux (µg/min)	SE	R^2^	*n*
SNV Natamycin	1.398·10^−6^	0.8663·10^−7^	0.293	0.018	0.981	0.163
HAV Natamycin	1.010·10^−6^	0.5985·10^−7^	0.212	0.012	0.965	0.194
LNV Natamycin	1.810·10^−6^	0.8780·10^−7^	0.380	0.018	0.998	0.053
Natacyn^®^	0.050·10^−6^	0.0612·10^−7^	0.075	0.009	0.971	0.629
SNV Voriconazole	4.52·10^−6^	2.722·10^−7^	1.358	0.081	0.981	0.163
HANV Voriconazole	3.110·10^−6^	1.800·10^−7^	0.931	0.053	0.982	0.160
LNV Voriconazole	4.145·10^−6^	1.606·10^−7^	1.243	0.048	0.998	0.064

**Table 8 pharmaceutics-15-00035-t008:** Pharmacokinetic parameters (*K*, *t_1/2_*, AUC _0_^∞^, and MRT) obtained by fitting the formulation percentage remaining in the eye over time by PET imaging.

Formulations	*K* (Min^−1^)		t_1/2_ (Min)		AUC 0^∞^ (% × Min)		MRT (Min)		R^2^	Remaining Formulation at 75 Min (%)	
	Mean	SD	Mean	SD	Mean	SD	Mean	SD		Mean	SD
SNV	0.010	0.007	12.02	2.23	46.77	8.19	68.47	4.45	0.96	15.42	6.107
AHNV	0.009	0.005	15.74	1.62	60.93	23.72	70.26	20.99	0.98	22.53	11.27
LNV	0.011	0.001	16.94	3.80	43.13	4.22	53.41	4.67	0.93	21.85	8.77

**Table 9 pharmaceutics-15-00035-t009:** Inhibition zone diameters (mm) obtained in Kirby–Bauer Disc Diffusion Method.

Fungal Specie	C.A 90231	C.A 90028	AF	PL	FS
Formulation					
SV	72	58	81	68	85
SN	33–35	32	28	0	47
SNV	74–76	62	72	80	77
VFEND^®^	61	61	87	84	103
NTC^®^	10	14	12	0	26

## Data Availability

Not applicable.

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
