# Peer review of "Antifungal Combination Eye Drops for Fungal Keratitis Treatment"

_pharmaceutics, 2022, doi:10.3390/pharmaceutics15010035_

Round 1

Reviewer 1 Report

Dear authors,

Thank you for submitting this manuscript to be considered for publication in Pharmaceutics Journal.

I enjoyed reading the manuscript and I found the work is very interesting, the experimental work very well designed, and the manuscript was very well written.

Good luck in you publication

Best wishes

Ismaiel 

Author Response

Dear Ismaiel,

The authors sincerely thank Ismaiel for the positive review of the manuscript titled “Antifungal Combination Eye Drops for Fungal Keratitis Treatment” and appreciate the time and effort that the reviewer has dedicated to providing their valuable feedback on the manuscript.

Reviewer 2 Report

The aim of this work is to develop a new ophthalmic formulation with the combination of natamycin and voriconazole based on the need to develop an effective and non-invasive treatment for fungal keratitis. Parameters including solubility of natamycin and voriconazole in HPβCD, binding interaction between natamycin and HPβCD, biosafe of the formulation were investigated. Meanwhile, the anti-fungal efficiency of the formulation against various fungi was measured.

Fungal keratitis is an important eye infection, while the solubility of natamycin and voriconazole limits the treatment efficacy. This work has an organized study plan and presentation of results. The solubility of natamycin and voriconazole, and the mechanism behind based on investigations of solubility diagram and the molecules interaction were studied. Meanwhile, problems found in clinical application such as transparency, osmolality, pH and drug release of Natacyn® were considered and assessed. The authors adopted different methods/assays/tests to explore the properties of Natacyn®. Corneal permeability, vessel modifications, corneal muco-adhesiveness were investigated in-vitro or on ex vivo models. After these investigations, the anti-fungal properties were assessed with Kirby-Bauer method.

All in all, this work indicated a clear study on optimizing the formulation of two anti-fungal drugs. The conclusions were supported by the results provided in the paper. The findings in this work can inspire related anti-fungal keratitis studies.

The authors should consider the following points to further improve the manuscript:

1. Line 404, “across corneal membranes better than the individual inclusion complexes” has no reference.

2. Line 406, “residence time at the site of administration” has no reference.

3. To better display the relationship between drug concentrations, content in Table 3 can be presented as diagrams, and with more experimental data points.

4. The NMR spectra are not clear to see each peak in Figure 4.

5. The data in Table 8 can be presented as several histograms to better reflect the difference among various drugs.

6. Although the anti-fungal properties were provide through measuring the inhibition zone, the treatment efficacy of different formulations on ex-vivo model or in vivo models have not been investigated yet.

Author Response

Dear Reviewer 2,

The authors sincerely thank Reviewer 2 for the positive review of the manuscript titled “Antifungal Combination Eye Drops for Fungal Keratitis Treatment” and appreciate the time and effort that Reviewer 2 has dedicated to providing his/her valuable feedback on the manuscript. A revised version of the text was submitted, addressing the concerns/doubts raised. All the new changes incorporated into the manuscript were highlighted in yellow.

Please, find below the point-by-point responses to the comments and concerns brought up by the Reviewer 2.

Reviewer 2

  1. Line 404, “across corneal membranes better than the individual inclusion complexes” has no reference.
  2. Line 406, “residence time at the site of administration” has no reference.

The authors sincerely thank the reviewer for their detailed comments and suggestions for the manuscript. The reference of line 404 and line 406 is reference [49] (Muankaew, C.; Saokham, P.; Jansook, P.; Loftsson, T. Self-Assembly of Cyclodextrin Complexes: Detection, Obsta-cles and Benefits. Die Pharmazie - An International Journal of Pharmaceutical Sciences 2020, 75, 307–312, doi:10.1691/ph.2020.0405.) that was in line 402. We moved the reference to line 406.

  1. To better display the relationship between drug concentrations, content in Table 3 can be presented as diagrams, and with more experimental data points.

Thank you for the comment. Table 3 shows the interaction parameters of the inclusion complexes calculated from solubility diagrams (Figure 1). When writing the manuscript, authors considered that a table was the simpler way to show these data as authors don’t have more experimental data points apart from those in Figure 1. If Reviewer 2 still considers that a diagram would simplify the shown data, please, specify which type of diagram should the authors use.

  1. The NMR spectra are not clear to see each peak in Figure 4.

Thank you for the comment. The arrangement of Figure 4 has been changed to make the image clearer.

  1. The data in Table 8 can be presented as several histograms to better reflect the difference among various drugs.

Thank you for your appreciation. Table 8 has been presented as six different histograms.

  1. Although the anti-fungal properties were provide through measuring the inhibition zone, the treatment efficacy of different formulations on ex-vivo model or in vivo models have not been investigated yet.

Thank you for your comment. The antifungal efficacy of the different formulations will be studied in the future. This new study will include new in vitro, ex vivo and in vivo efficacy methods. Authors have included a new sentence in lines 748-749 to clarify this. Also, in vivo assays in infected animals need for special permissions and installations that we don’t currently have.

Reviewer 3 Report

Please, small details - abbreviations are sometimes not explained /e.g. in Abstract NMR_

Author Response

Dear Reviewer 3,

The authors sincerely thank Reviewer 3 for the positive review of the manuscript titled “Antifungal Combination Eye Drops for Fungal Keratitis Treatment” and appreciate the time and effort that Reviewer 3 has dedicated to providing his/her valuable feedback on the manuscript. A revised version of the text was submitted, addressing the concerns/doubts raised. All the new changes incorporated into the manuscript were highlighted in yellow.

Please, find below the point-by-point responses to the comments and concerns brought up by the Reviewer 3.

  1. Please, small details - abbreviations are sometimes not explained /e.g. in Abstract NMR.

Thanks for your appreciation, NMR abbreviation has been explained in Abstract.